# Measuring Dietary Intake of Pregnant Women Post-Bariatric Surgery: Do Women Meet Recommendations?

**DOI:** 10.3390/nu17020285

**Published:** 2025-01-14

**Authors:** Taylor M. Guthrie, Sandra Lee, Alka Kothari, Sailesh Kumar, Helen Truby, Susan de Jersey

**Affiliations:** 1Faculty of Health, Medicine & Behavioral Sciences, University of Queensland, St Lucia, QLD 4072, Australia; alka.kothari@health.qld.gov.au (A.K.); sailesh.kumar@mater.uq.edu.au (S.K.); h.truby@uq.edu.au (H.T.); susan.dejersey@health.qld.gov.au (S.d.J.); 2Dietetics and Foodservices, Royal Brisbane Women’s Hospital, Herston, QLD 4006, Australia; 3Maternity Services, Caboolture Hospital, Caboolture, QLD 4510, Australia; sandra.lee@health.qld.gov.au; 4Obstetrics and Gynecology, Redcliffe Hospital, Redcliffe, QLD 4020, Australia; 5Mater Research Institute, University of Queensland, South Brisbane, QLD 4072, Australia

**Keywords:** dietary intake, pregnancy, bariatric surgery, micronutrients, deficiency, supplementation

## Abstract

**Background:** Bariatric surgery is increasingly offered to women of childbearing age and significantly reduces food intake and nutrient absorption. During pregnancy, associated risks, including micronutrient deficiency, are accentuated. This study describes maternal dietary intake and adherence to dietary recommendations in pregnant women with a history of bariatric surgery. **Methods:** Women aged 18–45 with singleton pregnancies post-bariatric surgery were recruited at <23 weeks of gestation and followed until birth. Dietary intake was measured using three non-consecutive 24 h recalls at enrolment and at 28 and 36 weeks using the standardized tool ASA24-Australia. Micronutrient supplementation dose and adherence was reported using the Brief Medication Adherence Questionnaire. Mean macronutrient intake was calculated from all diet recalls. Micronutrient intake was determined from diet recalls and from supplementation. Intake was compared to the recommended daily intakes for pregnancy. **Results:** Sixty-three women participated in the study. The participants met 65 ± 17.3% (mean ± SD) of estimated energy requirements, 53(23)% (median(IQR)) of fiber requirements, and exceeded fat and saturated fat recommendations. Dietary intake levels of iron, folate, zinc, calcium, and vitamin A were below recommended levels. Gastric bypass recipients consumed significantly less folate (*p* = 0.008), vitamin A (*p* = 0.035), and vitamin E (*p* = 0.027) than women post-gastric sleeve or gastric band. Multivitamins were used by 80% (n = 55) of participants at study enrolment, which increased their mean intake of all micronutrients to meet recommendations. **Conclusions:** Women who conceive post-bariatric surgery may require targeted support to meet the recommended nutrient intake. Micronutrient supplementation enables women to meet nutrient recommendations for pregnancy and is particularly important for gastric bypass recipients.

## 1. Introduction

In developed countries, 40 to 70% of women of childbearing age are classified as overweight or obese [1,2,3], a factor associated with infertility [4] and other serious pregnancy-related complications [5], which have lifelong health impacts [6]. There is good evidence that appropriate weight reduction before conception significantly improves fertility and perinatal outcomes in women classified as overweight or obese [7,8]. As bariatric surgery is now recognized as a highly effective weight-loss strategy [9,10,11,12], increasing numbers of women of childbearing age are requesting this intervention [13,14]. Following bariatric surgery, individuals typically lose 20 to 35% of their body weight through multiple mechanisms including decreased food intake, hypoabsorption of nutrients, and altered hormonal appetite control mechanisms [15,16,17]. Although dietary intake is profoundly changed post-operatively, surgery does not necessarily improve diet quality [18] as this depends on how well overall food intake aligns with dietary guidelines [19]. A high-quality diet should include items from the five food groups: fruits, vegetables, high-fiber grains, low-fat dairy, and lean meats or alternatives and should be aligned with the minimization of discretionary foods typically high in saturated fat, added sugars, and sodium. A high-quality diet provides an optimal balance of macro- and micronutrients for overall health and wellbeing and reduces chronic disease risk [18,19].

Side effects of bariatric surgery may include nausea, vomiting, gastroesophageal reflux, and bowel dysfunction, which can heavily influence subsequent dietary intake [20]. Dumping syndrome, which occurs in approximately 40% of cases, also produces significant post-prandial symptoms (nausea, vomiting, syncope, diarrhea, and hypoglycemia) [21,22]. The combined impact of post-operative symptoms and reduced dietary intake can result in post-operative micronutrient deficiency (particularly iron, folate, copper, calcium, thiamine, zinc, and vitamins A, B12, and D) [15,18,23,24]. To address this concern, current guidelines [15,25] recommend lifelong vitamin and mineral supplementation for all bariatric surgery patients.

As dietary intake, weight, and micronutrient status are key determinants of maternal and child health outcomes [26], there are concerns that the increased nutritional requirements during pregnancy may not be adequately addressed after bariatric surgery. Dietary energy consumption is a key determinant of gestational weight gain [26], and current Institute of Medicine (IOM) gestational weight gain recommendations are based on pre-pregnancy body mass index (BMI) [27]. Indeed, regardless of pre-pregnancy BMI, inadequate gestational weight gain is associated with increased risks of preterm birth and small-for-gestational-age (SGA) babies [26,27,28]. Similarly, higher SGA [29,30,31,32,33,34,35,36] and preterm birth rates [29,30,31,33,34] have also been observed in women who conceive post-bariatric surgery.

As pregnant women post-bariatric surgery are at significant risk of iron, folate, and vitamins A, B12, D, and K deficiency, Shawe and colleagues [16] developed clinical consensus guidelines recommending that these women should receive micronutrient supplementation and regular dietetic care to help them achieve appropriate gestational weight gain. However, the evidence for these recommendations is limited—a recent systematic review [37] identified only one study that reported on both dietary intake and the use of micronutrient supplements during pregnancy following bariatric surgery [38]. Given this background of limited evidence, the aim of this study was to measure dietary intake during pregnancy in women who have had bariatric surgery and compare their dietary intake to nutrient reference values for pregnancy. A secondary aim was to describe the use and dose of micronutrient supplements throughout pregnancy.

## 2. Materials and Methods

### 2.1. Study Design and Setting

This was a prospective cohort study of pregnant women with a history of any type of bariatric surgery from four hospitals across Queensland, Australia. Ethical approvals were granted by the Metro North Hospital and Health Service Human Research and Ethical Committee (EC00168) and the University of Queensland Human Ethics Committee. The study protocol was registered with the Australian and New Zealand Clinical Trials Registry (registration number ACTRN12623000495628). At all recruiting hospitals, no specific changes to routine pregnancy care were made [16,39]. Participating women were recruited before week 23 of gestation and remained in the study until the birth of their infant. Eligibility criteria included: women aged 18–45 years with a history of bariatric surgery (gastric band, gastric sleeve, gastric bypass, or revisional procedure) who were able to provide informed consent. Exclusion criteria included: women with multiple pregnancy or any conditions that could impact micronutrient absorption or metabolism (such as renal failure, inflammatory bowel disease, or type 1 diabetes mellitus). Eligible women were provided information about the study by either midwives, dietitians, obstetricians, or obstetric physicians in person or by text message, and those who expressed interest in participating were subsequently counselled in detail before written consent was obtained. Participant information and consent forms were distributed using a secure Redcap [40,41] survey sent by email.

### 2.2. Data Collection Methods

Online questionnaires were sent by email using a secure Redcap link [40,41] at study enrolment (<23 weeks) and at 28 and 36 weeks. Information regarding maternal age, estimated date of confinement, ethnicity, educational attainment, marital and employment status, height, pre-surgery weight, pre-pregnancy weight, bariatric surgery procedure received and date of surgery, current weight, details regarding micronutrient supplementation, dietary intake, and physical activity levels were collected at enrolment and at 28 and 36 weeks. The use of and adherence to micronutrient supplements (including multivitamins and single nutrient supplements) were collected using the Brief Medication Adherence Questionnaire, which also quantifies missed doses across the week [42]. Physical activity was measured using the International Physical Activity Questionnaire (IPAQ) short form [43], and dietary intake was measured using three non-consecutive 24 h recalls at each timepoint using a standardized tool (ASA24-Australia) [44]. The ASA24-Australia uses an automated multiple-pass method that quantifies dietary intake using the Australian Food, Supplement, and Nutrient Database 2011–2013 and provides the total macronutrient and micronutrient consumption for each recall. Diet recalls were primarily conducted over the telephone by a single researcher who entered reported food intake into the ASA24-Australia online tool. This researcher (T.G.) was an experienced dietitian with extensive food knowledge and used household measurements to identify portion sizes. Additional data collected from participants’ medical records after birth included parity and details of dietitian care and receipt of intramuscular vitamin B12 replacement during pregnancy.

### 2.3. Determination of Dietary Intake and Nutrient Requirements

The mean intake of all macronutrients and micronutrients was determined from recalls collected at each data collection point and an average calculated across all three recalls. Energy intake was expressed in kilojoules (kJ) per day and per kilogram (kJ/kg). Energy requirements for each participant were estimated based on the Dietary Reference Intake method [45] for the second and third trimesters of pregnancy, considering physical activity level (PAL), age, height, and pre-pregnancy BMI. Participant responses to the IPAQ questionnaire were categorized as being low, moderate, or high in accordance with the IPAQ guidelines [46]. This was used to determine the appropriate PAL for energy requirement calculations:
For participants who scored “low” on the IPAQ, the equation for an “inactive” PAL was applied.For those with a “moderate” activity level, the “active” PAL equation was used.Finally, for participants with a “high” IPAQ score, the equation for a “very active” PAL was applied.

Descriptive statistics were used to compare dietary intake to nutrient recommendations. The proportion of energy intake from protein, carbohydrates, fat, and saturated fat was compared to the acceptable macronutrient distribution range [47]. Protein intake was further compared to the recommendations made by Shawe et al. [16] for pregnancy after bariatric surgery of at least 60 g/day. The intake levels of fiber, iron, calcium, selenium, zinc, folate, and vitamins A, B12, and E were compared to the Australian Recommended Daily Intake (RDI) or Adequate Intake (AI) for pregnancy [47]. For the purpose of this study, diet quality was defined according to adherence to the acceptable macronutrient distribution range and whether the recommended micronutrient intakes for pregnancy were met.

### 2.4. Determination of Micronutrient Supplementation

Micronutrient supplementation was calculated for each of the three data collection points. Participants provided the name and brand of multivitamin multimineral supplement used, as well as any individual supplements they had taken in the previous two weeks. Average daily dose of iron, calcium, selenium, zinc, folic acid, and vitamins A, B12, and E was then calculated by multiplying the supplement dose by the number of tablets taken and days taken and subtracting missed doses across the week. Total micronutrient intake was calculated by combining the average daily intake of micronutrients with that from dietary sources.

### 2.5. Data Analysis

Approximate date of conception was calculated by subtracting 280 days from the estimated date of confinement provided by the attending medical practitioner. This was used to determine the maternal age at conception and the bariatric-surgery-to-conception interval. Gestational weight gain was calculated by subtracting pre-pregnancy weight from weight at 36 weeks and was compared to the IOM recommendations according to pre-pregnancy BMI [27]. Nutrient intake was reported as a mean with standard deviation (SD) (normally distributed variables) or median with interquartile range (IQR) (non-normally distributed variables) and as a percentage of the Australian nutrient reference values [47]. Change in nutrient intake from enrolment to 28 and to 36 weeks was determined using a repeated measures ANOVA and paired samples *T*-test (normally distributed data), applying a Friedman’s or a Mann–Whitney test as appropriate (non-normally distributed data). Macronutrient intake was compared between women with recommended, below-recommended, and above-recommended gestational weight gain using a one-way ANOVA (normally distributed variables) or Kruskal–Wallis H test (non-normally distributed variables). Any impact of the surgery-to-conception interval (<12 and ≥12 months) and surgery type (non-hypoabsorptive surgeries: gastric band and gastric sleeve, hypoabsorptive surgeries: gastric bypass) on nutrient intake was explored by applying an independent samples *t*-test (normally distributed variables) or Mann–Whitney U test for non-normally distributed variables. A Pearson correlation was used to examine how the number of dietitian appointments and gestation at first appointment related to gestational weight gain and macronutrient and micronutrient intake. A *p*-value < 0.05 was considered statistically significant.

## 3. Results

A total of 74 women consented to participate, with 62 remaining in the study until the birth of their child (see Figure 1 for participant flow). Participant demographics are summarized in Table 1. Participants completed a median (IQR) of six (four) diet recalls during the study (Appendix A). Physical activity level throughout pregnancy is provided in Table 2.

### 3.1. Macronutrient Intake

Mean energy intake (7462 ± 1831 kJ) met 65 ± 17% of estimated energy requirements for pregnancy across the study period. Participants consumed a median of 73 (21.3) g of protein and 0.84 (0.4) g per kg according to pre-pregnancy weight. The majority (87%, n = 60) of participants met or exceeded the minimum recommended 60 g per day protein intake during pregnancy, with protein intake providing 17 (3.5)% of total energy intake. Carbohydrate intake provided 42 ± 6.1% of energy intake; however, fiber intake was consistently below the AI. The mean proportion of energy from fat and saturated fat intake exceeded recommendations (Table 3). There was no significant change over time in the proportion of macronutrients between enrollment and weeks 28 and 36 (n = 35, *p* = 0.119 − 0.819) or between enrollment and week 28 (n = 55, *p* = 0.111 − 0.916). Daily macronutrient intake was similar between women who conceived at <12 and ≥12 months after surgery (*p* = 0.554 − 0.986). There were no significant differences in micronutrient intake between those who had received hypoabsorptive or non-hypoabsorptive surgery (*p* = 0.995 − 0.068). Women with recommended, below-recommended, and above-recommended gestational weight gain, as defined by the IOM recommendations, consumed similar proportions of their total estimated energy requirements (*p* = 0.074), and consumed similar proportions of protein (*p* = 0.297), fat (*p* = 0.288), saturated fat (*p* = 0.178), and carbohydrates (*p* = 0.178) as percentages of total energy intake.

### 3.2. Micronutrient Intake

The median consumption levels of iron, zinc, calcium, folate, and vitamins A and B12 from dietary sources fell below recommendations. Micronutrient intake from dietary sources was similar among those with different surgery-to-conception intervals (*p* = 0.645 − 0.969). Women who had undergone hypoabsorptive bariatric surgery consumed significantly less folate (*p* = 0.008), vitamin A (*p* = 0.035), and vitamin E (*p* = 0.027) than participants who had undergone non-hypoabsorptive surgery types (Figure 2). Multivitamin supplements were used by 80% of participants at study enrollment, with 74% continuing throughout pregnancy (Figure 3). Fewer participants used single nutrient supplements. The median number of missed supplement doses was zero for multivitamins and individual supplements across pregnancy, indicating high adherence to supplementation. Adherence to supplementation did vary between participants, however, and there was a trend towards more missed doses in early pregnancy (Table 4). With supplementation, the median consumption of all micronutrients met or exceeded the RDI or AI (Table 5).

### 3.3. Dietitian Care

In total, 86% of participants saw a dietitian during pregnancy. The median (IQR) gestation of first appointment was 17 (5) completed weeks, and women attended a median (IQR) of four (two) appointments. The gestation at the time of the first dietetic appointment was not correlated with gestational weight gain (*p* = 0.479), macronutrient intake (*p* = 0.484 − 0.957), or micronutrient consumption from dietary sources alone (*p* = 0.361 − 0.998) or when combining intake from dietary sources and supplementation (*p* = 0.062 − 0.919). The number of dietetic appointments was negatively correlated with gestational weight gain (R = −0.317, *p* = 0.017) and positively correlated with carbohydrate intake (R = 0.274, *p* = 0.036) (Appendix A). Fat (R = −0.251, *p* = 0.055), saturated fat (R = −0.255, *p* = 0.051), protein (R = −0.078, *p* = 0.556), and micronutrient intake (*p* = 0.150−0.896) from dietary sources were not significantly correlated with the number of appointments attended with a dietitian. Micronutrient intake when combining dietary sources and supplements was correlated with the number of dietitian appointments for vitamin A (R = 0.399, *p* = 0.002) and vitamin B12 (R = −0.273, *p* = 0.036). Intramuscular vitamin B12 was received by 22% (n = 15) participants. 

## 4. Discussion

The results of this study show that the dietary intake of pregnant women post-bariatric surgery is high in fat and saturated fat and low in fiber, iron, calcium, zinc, vitamin A, and folate. We also show that micronutrient supplementation is used by a large proportion of women throughout pregnancy, with variable adherence. Greater access to dietetic care was also shown to have a small but significant relationship with improved gestational weight gain, carbohydrate intake, and vitamin A and B12 consumption from diet and supplements.

Despite the recommendation for post-bariatric surgery women to receive dietetic care during pregnancy, there is limited available evidence to guide medical nutrition therapy [16]. Little is known about the nutrient requirements for this group, and determining macronutrient needs primarily relies on accurately estimating energy (kilojoule) requirements. In this study, we applied the best methods available for measuring food intake (ASA24 method), with the additional benefit of the dietitian adopting a 24 h recall to support accurate reporting [48]. However, methods for estimating energy requirements are primarily developed based on women with a pre-pregnancy BMI between 20 kg/m^2^ and 27 kg/m^2^, [49,50] whereas participants in this study had a median pre-pregnancy BMI of 30 kg/m^2^. In this study, participants met an average of 65% of their estimated energy requirements; however, two thirds achieved a gestational weight gain within or above recommendations. This suggests that current predictors of energy expenditure may overestimate true requirements for some women—a trend also noted in studies involving pregnant women without bariatric surgery [49]. Research on energy expenditure during pregnancy has also shown considerable individual variation [51]. This underscores the need to individualize energy and macronutrient recommendations during pregnancy for these women.

Beyond energy consumption, macronutrient balance also contributes to maternal and offspring health outcomes. High fat and saturated fat intake are thought to be implicated in maternal hyperglycemia and insulin resistance [52], as well as offspring fat mass [53]. A study by Khaire et al. [54] reported a maternal fat intake of over 35% of energy intake being correlated with increased BMI and waist circumference among male offspring. Participants in the present study consumed 40% of their energy from fat. Some studies have reported greater BMI z-scores [55] and decreased height [56] in the children of mothers who have undergone bariatric surgery; however, these have not examined the role of maternal dietary intake. As fat increases postprandial satiety, it may limit the intake of other macronutrients, for example, carbohydrates and fiber [26]. Participants in this study reported levels of carbohydrate intake below (42% energy intake) the acceptable macronutrient distribution range (45–65% energy intake) and low fiber intake (53% of the AI). This dietary pattern has been associated with inadequate micronutrient intake during pregnancy, preterm birth and small-for-gestational-age offspring [57]. Conversely, evidence also suggests that insufficient fat and vitamin E contribute to preterm birth risk, with Ref. [52] highlighting a need to balance nutrient intake by optimizing food choices that are appropriate and suitable for those who live with bariatric procedures and their consequences. Existing recommendations suggest women eat at least 60 g of protein per day during pregnancy [16]. The majority of participants in this study met this recommendation and yet still had significant nutritional shortfalls, suggesting a need to address diet quality holistically rather than focus on individual nutrients.

Sub-optimal micronutrient intake potentially leads to micronutrient deficiency during pregnancy, is implicated in the development of several perinatal complications [58,59,60], and may impact offspring nutrient status in early life [59]. In particular, after bariatric surgery, iron, folate, and vitamin A, B12, D, and K deficiency have been reported [37]. In a systematic review including only women who adhered to micronutrient supplementation, 15 to 75% experienced iron deficiency and 30 to 91% developed vitamin A deficiency [37]. These studies did not report dietary intake. However, the present study suggests that inadequate dietary intake of these nutrients may contribute to deficiency risk. The inadequate consumption of calcium, zinc, iron, and vitamin A observed in our sample have also been correlated with preterm birth and small-for-gestational-age neonates [26], which are observed more frequently in pregnancies post-bariatric surgery [16]. The women in this longitudinal study concerningly met only 61% of folate requirements from dietary sources, with particularly low intake noted among women who had bypass procedures, whose absorption is further compromised [15]. Low folate intake is strongly associated with risk of neural tube defects [26]. While some meta-analyses have reported higher risks of neural tube defects following bariatric surgery [29], findings have been inconsistent [33]. In this study, many participants consumed additional folic acid through multivitamins (74 to 80%) or folic acid supplements (13 to 25%), which contributed to their total folate intake falling within recommended levels [16]. Whilst these findings are based on a small sample size, they may help explain the mixed findings in prior research on neural tube defects post-bariatric surgery and highlight the importance of supplementation, particularly for gastric bypass recipients.

This study is the first to compare maternal dietary intake to existing nutrient reference values for pregnancy over time. It is important to acknowledge that dietary recommendations have been developed using data on pregnant women without a history of bariatric surgery. Due to hypoabsorption resulting from bariatric surgery, post-operatively, women may need to regularly exceed the RDI to attenuate their risk of deficiency [15,16]. Current supplementation guidelines recommend women exceed the RDI for iron, calcium, zinc, selenium, copper, folic acid, thiamine, and vitamins A, D, E, and K through additional micronutrient supplements [16,39]. It is concerning that women in this sample are falling short of the RDI for several micronutrients given they may have even higher needs to prevent nutrient deficiency. Micronutrient supplementation is not without drawbacks, as it is associated with pill burden, side effects, and costs for women [61]. However, this study suggests that additional micronutrient supplementation is essential for these women. Interventions that improve overall diet quality may address both macronutrient and micronutrient consumption and improve perinatal outcomes and therefore warrant investigation.

Individualized care from an experienced dietitian is recommended for women who become pregnant following bariatric surgery, both while planning and during the pregnancy [16]. However, access to such care varies widely. A survey of Australian antenatal care professionals found that only 50% refer women with a history of bariatric surgery to a dietitian [62]. Encouragingly, 86% of participants in this study consulted a dietitian during pregnancy; however, most did not start until the second trimester. Poor diet quality has been noted as a concern among non-pregnant bariatric surgery recipients [18], and this study suggests that these challenges persist during pregnancy. Consumers have expressed a strong desire for support with dietary intake both prior to and throughout pregnancy [63,64]. In this study, an increase in dietetic appointments correlated with some improvements in macronutrient and micronutrient intake, although statistically, this relationship was small. Given the lack of evidence-based nutrition recommendations for this population, it is unclear what intervention advice women in our study received during those dietetic appointments. The knowledge and confidence of dietitians providing care to pregnant women post-bariatric surgery has not been investigated; however, studies including other antenatal health professionals have reported knowledge gaps in the care of this increasingly large population [62,65]. This may explain the relatively small influence on nutrient intake noted in our sample. Therefore, while early and frequent individualized dietetic intervention could be viewed as a crucial link between improving dietary intake and maternal gestational weight gain, further research is needed to understand how dietetic interventions are most effective in enhancing dietary intake during pregnancy. Subsequently, the development of evidence-based guidelines that are co-designed with consumers may further enhance the acceptability of dietary advice.

Our study had several strengths and limitations. We employed robust methodology to capture dietary intake and supplement use by using validated tools and repeating measures several times throughout pregnancy. Dietary intake and supplement use was also collected by an experienced researcher whose expertise in the field assisted in the accurate determination of food intake and supplement use. Recall bias may have impacted data collection; however, the protocol aimed to limit this by collecting data throughout pregnancy. The fact that nutrient intake consistently fell below recommendations may also be caused by underreporting of food intake, which is common in dietary intake studies, particularly those including people living with obesity [66]. However, the study protocol aimed to prevent misreporting through the use of a dietitian-assisted multiple-pass 24 h recall method alongside a best-practice automated dietary analysis tool. With no thresholds to identify misreporting, the data havee been reported with no adjustments to provide a contemporary picture of dietary intake and change over the course of a pregnancy in women who have received bariatric surgery [67]. Other potential limitations of this study include small sample size, non-response bias, and under representation of women who received bypass procedures compared to international data. This does, however, reflect Australian bariatric surgery trends [14].

## 5. Conclusions

This study identified significant gaps in macronutrient distribution and micronutrient intake among pregnant women with a history of bariatric surgery, despite meeting suggested protein intake recommendations [16]. Gestational weight gain data suggest that women in this study largely met the increased energy (kilojoule) requirements for pregnancy; however, they appeared to consume foods lower in nutrient value, which likely explains the low fiber and micronutrient intake. Dietary intake of micronutrients was particularly low among gastric bypass recipients, who face more-significant deficiency risk due to nutrient hypoabsorption [15]. This suggests that adverse perinatal outcomes may not reflect inadequate energy or protein intake but rather poor-quality diet in women who conceive post-bariatric surgery. These findings underscore the necessity of targeted micronutrient supplementation for women who have undergone bariatric surgery, particularly those with gastric bypass. Improving diet quality may improve macronutrient distribution and micronutrient intake for these women, and potentially maternal and offspring health outcomes [26,57]. Given that dietetic interventions have some influence on nutrient intake and gestational weight gain, early engagement with experienced dietitians should be a key priority for health services as this population continues to grow.

## Figures and Tables

**Figure 1 nutrients-17-00285-f001:**
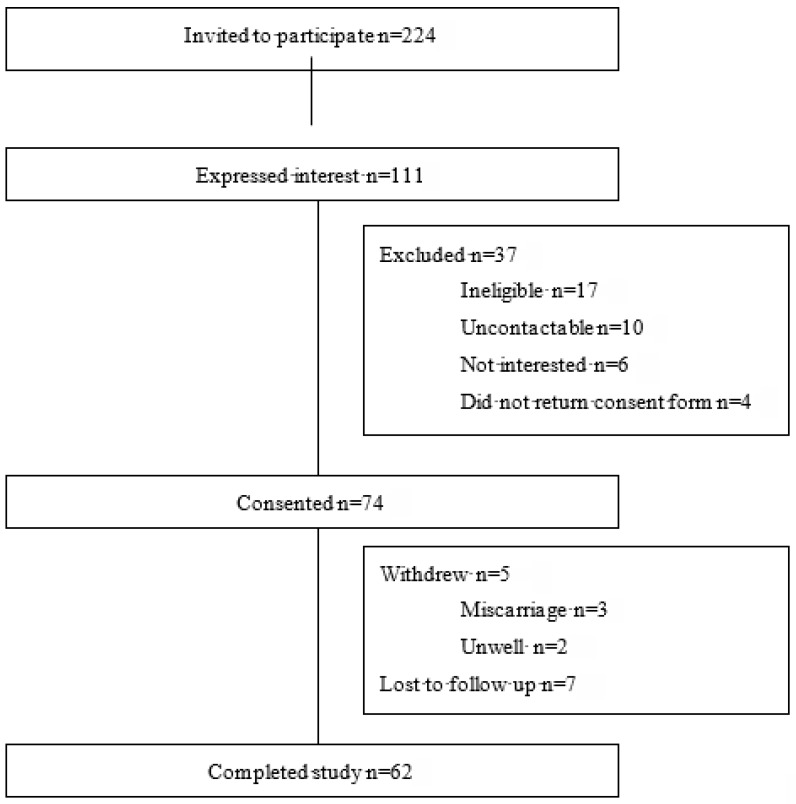
Participant Flow.

**Figure 2 nutrients-17-00285-f002:**
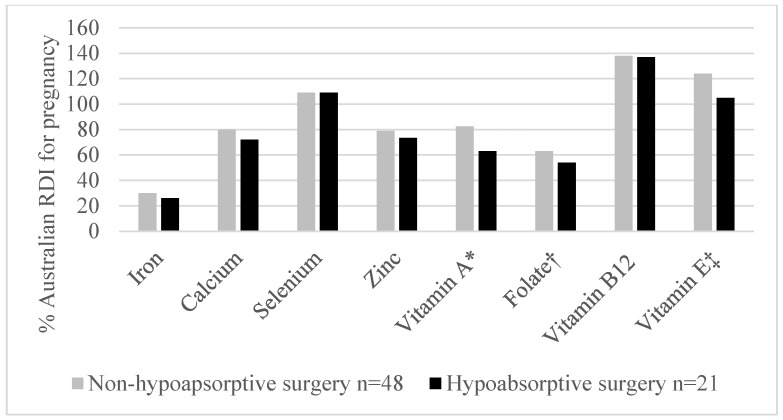
Impact of bariatric surgery type on micronutrient intake from dietary sources. * *p* = 0.035, † = *p* = 0.008, ‡ *p* = 0.027.

**Figure 3 nutrients-17-00285-f003:**
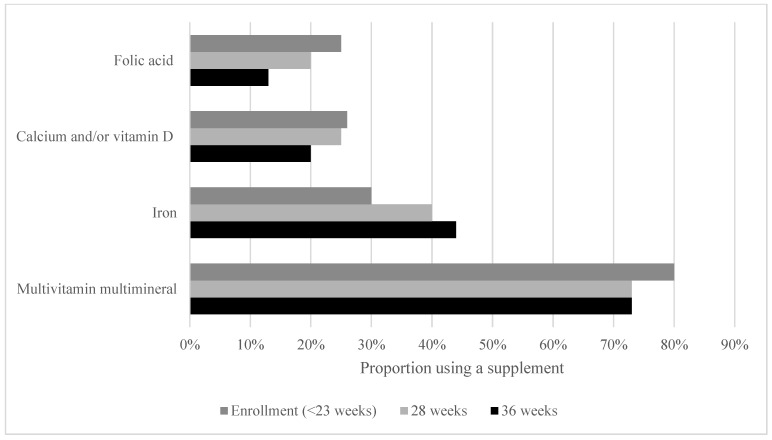
Micronutrient supplement use throughout pregnancy.

**Table 1 nutrients-17-00285-t001:** Participant demographics.

Variable	Mean ± SD	Range
Maternal age at conception (years)	31.2 ± 4.8	22–42
Gravida status	3 (2) *	1–10
Parity	1 (2) *	0–4
Pre-bariatric surgery BMI (kg/m^2^)	45.1 (12.7) *	32.7–73.1
Pre-pregnancy BMI (kg/m^2^) BMI Category, % (n=) BMI < 18.5 BMI 18.5–24.9 BMI 25–29.9 BMI > 30	30.4 (7.3) * 6 (n = 4) 13 (n = 9) 26 (n = 18) 55 (n = 38)	18.4–50.4
Gestational weight gain (kg) Adherence to IOM recommendations, %(n=) † Below recommendations Within recommendations Above recommendations	7.4 ± 7.3 37 (n = 18) 31 (n = 15) 33 (n = 16)	−12–22
Bariatric-surgery-to-conception interval (months) <12 months, % (n=)	30 (51) * 19 (n = 13)	0–186
Bariatric surgery type, %, (n=) Gastric band Gastric sleeve Gastric bypass Revisional surgery % (n=)	1 (n = 1) 68 (n = 47) 30 (n = 21) 7 (n = 5)	
Ethnicity, %(n=) ‡ Caucasian Aboriginal or Torres Strait Islander Pacific Islander Other	78 (n = 53) 10 (n = 7) 3 (n = 2) 9 (n = 6)	
Marital status, %(n=) ‡ Married or de facto Never married Divorced or separated	65 (n = 44) 27 (n = 18) 9 (n = 6)	
Education attainment, %(n=) ‡ Postgraduate qualification Undergraduate degree Trade, technical certificate or diploma Completed grade 12 Completed grade 10 Other	6 (n = 4) 18 (n = 12) 38 (n = 26) 28 (n = 19) 9 (n = 6) 6 (n = 1)	

* Median (IQR), † total n = 49, ‡ total n = 68 (n = 1 missing).

**Table 2 nutrients-17-00285-t002:** Physical activity level throughout pregnancy.

Physical Activity Level	Enrolment (<Week 23)	Week 28	Week 36
Low activity, % (n=)	41 (n = 28)	40 (n = 26)	55 (n = 30)
Moderate activity, % (n=)	48 (n = 33)	54 (n = 35)	38 (n = 21)
High activity, % (n=)	12 (n = 8)	6 (n = 4)	7 (n = 4)
Total n=	n = 69	n = 65	n = 55

**Table 3 nutrients-17-00285-t003:** Macronutrient consumption of pregnant women post-bariatric surgery across all data collection points.

Macronutrient	Mean ± SD	Macronutrient Proportion (%) of Total Energy Intake	Recommended Daily Intake
Energy (kJ) kJ/kg *	7495 ± 1816 89.2 ± 31.1	-	11,685 ± 1115 ‡
Protein (g) g/kg *	73.0 (21.3) † 0.84 (0.41) †	17 (3.5) †	15% energy intake §
Carbohydrate (g)	184.9 ± 52.7	42 ± 6.1	45–65% energy intake §
Fat (g)	80.0 ± 23.8	40 ± 5.2	<30% energy intake §
Saturated fat (g)	33.4 ± 10.2	17 ± 2.8	<10% energy intake §

* Pre-pregnancy weight, † median (IQR), ‡ dietary reference values for energy requirements during pregnancy [45], § acceptable macronutrient distribution range [47].

**Table 4 nutrients-17-00285-t004:** Adherence to micronutrient supplementation throughout pregnancy.

Supplement Type	Enrolment (<Week 23) % Doses Missed/Week		Week 28 % Doses Missed/Week		Week 36 % Doses Missed/Week	
	Median (IQR)	Range	Median (IQR)	Range	Median (IQR)	Range
**Multivitamin multimineral**	0 (14)	0–86%	0 (14)	0–57%	0 (19)	0–75%
**Iron only**	0 (27)	0–100%	0 (23)	0–67%	0 (14.3)	0–57%
**Calcium and/or vitamin D**	0 (18.5)	0–86%	0 (26.8)	0–75%	0 (0)	0–29%
**Folic acid only**	0 (28.6)	0–75%	0 (22.6)	0–75%	0 (0)	0–40%

**Table 5 nutrients-17-00285-t005:** Key micronutrient and fiber intake from dietary sources with and without supplements across all data collection points.

Micronutrient	Dietary intake per Day		Intake per Day from Dietary Sources Plus Supplements		Australian RDI for Pregnancy
	Median (IQR)	% RDI	Median (IQR)	% RDI	
**Fiber (g)**	15 (6.4)	53 (23) †	NA	NA	28 g †
Iron (mg)	8.0 (3.0)	29 (11.2)	119 (182.3)	442 (675)	27 mg
Calcium (mg)	762 (305)	76 (30.5)	1077 (508.8)	108 (50.9)	1000 mg
Selenium (mcg)	71 (18.5)	109 (28.5)	194 ± 89.3 *	299 ± 196	65 mcg
Zinc (mg)	9 ± 2.4 *	78 ± 22.2 *	35 (26.8)	314 (243)	11 mg
Vitamin A (IU)	2082 (1411.6)	78 (52.9)	2954 (2435.9)	111 (91.3)	2667 IU
Folate (mcg)	365 (139.2)	61 (23.2)	2146 (1470.4)	358 (245)	600 mcg
Vitamin B12 (mcg)	4 (1.4)	137 (53.3)	12 (37.5)	448.6 (1443.0)	2.6 mcg
Vitamin E (mcg)	8 (2.9)	118 (41.8)	22 (25.7)	315 (367.7) †	7 mcg †

* Mean ± SD, † adequate intake [47].

## Data Availability

The data presented in this study are available on request from the corresponding author due to ethical reason.

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
