# Peer review of "Measuring Dietary Intake of Pregnant Women Post-Bariatric Surgery: Do Women Meet Recommendations?"

_nutrients, 2025, doi:10.3390/nu17020285_

Round 1

Reviewer 1 Report

Comments and Suggestions for Authors

I read the manuscript on the dietary intake of post-bariatric women during pregnancy with great interest.

In general, the authors employ a robust methodology and conduct a thorough statistical analysis. However, I am not sure whether the question posed in the article's title is relevant or if their research is appropriately designed to address it ("is multivitamin supplement essential?").  To effectively address this, they should have measured blood levels of the relevant nutrients and correlated them with patients' adherence to supplement intake. Simply relying on a questionnaire does not adequately answer the question, especially given the inherent limitations of this method. In this regard, they should explicitly acknowledge 'recall bias' and 'non-response bias' in the limitations section. Similarly, the response rate is reasonable, with 62 patients out of 224 initially invited to participate.

Here also some remarks regarding appropriate terminology: 1) The correct term is 'hypoabsorptive' procedures, not 'malabsorptive.' Malabsorption is a complication and represents a serious medical condition. Some metabolic bariatric procedures aim to reduce calorie absorption, but under no circumstances do they—or should they—intend to cause malabsorption. 2) Very few, if any, procedures are purely restrictive or hypoabsorptive. Even Roux-en-Y gastric bypass includes a significant restrictive component due to the small size of the gastric pouch. Similarly, procedures historically considered restrictive, such as sleeve gastrectomy, have a substantial hormonal component, as evidenced by the decrease in ghrelin levels following the removal of the gastric fundus. It would be more appropriate to adopt the terms 'hypoabsorptive' and 'non-hypoabsorptive' procedures.

The text does not mention protein intake, despite the presence of the corresponding correlation diagram (Figure 4.1). Additionally, there is a discrepancy in the correlation coefficient for saturated fat intake, reported as -0.255 in the text but shown as 0.274 in the diagram. Overall, these diagrams may not be essential for the main body of the text, given the weak correlations (<0.30). It would be more appropriate to include them as supplementary material.

Thank you for giving me the opportunity to review your article.  I look forward to a constructive discussion.

Author Response

We are grateful to both reviewers for their comments and feel their advice has strengthened this manuscript. 

Please find attached a completed response to reviewer comments. 

Kind regards, 

Taylor Guthrie 

Reviewer 2 Report

Comments and Suggestions for Authors

Here are my suggestions and comments to improve your work:

1-Line 50, why don't the authors write about essential fatty acids, in pregnancy are pivotal?

2-line 128 could the author use PAL instead of the full term?

3-lines 130-135, please insert the 3 equations used for the three levels of PAL.

4-lines 138-139 explain why you used 60 g/day instead of 0.8-1 g pro kg of protein as recommended for pregnant women.

5- In supplementary materials (lines 195-196), please consider inserting a table of macronutrients in the 28-week and 36-week.

6-Table 4 why all the median values are 0? It seems to be present as a typo.

7-line 258, insert in the limitation recall 24 h because it is less accurate than the gold standard 7-day diary record.

Author Response

(The authors gave the same response as above.)

Round 2

Reviewer 1 Report

Comments and Suggestions for Authors

Change this: "non-hypoabsorptive surgeries: gastric band and gastric bypass, hypoabsorptive surgeries: gastric bypass" to this: "non-hypabsorptive surgeries: gastric band, sleeve gastrectomy; hypoabsorptive surgeries: gastric bypass (all types)".

The rest of my suggestions have been effectively addressed, you can proceed to publication.

Author Response

Thank you for your comments - this have been actioned. 

I have also changed Figure 2 to align with the new terminology of 'hypoabsorptive' and 'non-hypoabsorptive' surgery types. 

I have left Figure 4 in the manuscript however this need to be moved to Supplementary materials as per reviewer feedback from round 1. I have left a comment in the manuscript to this effect. 

Thank you.